```
1
2
```

## Title: New observations confirm the progressive acidification in the Mozambique Channel

Nicolas Metzl<sup>1</sup>, Claire Lo Monaco<sup>1</sup>, Aline Tribollet<sup>1</sup>, Jean-François Ternon<sup>2</sup>, Frédéric Chevallier<sup>3</sup>, Marion Gehlen<sup>3</sup>

- 1 Laboratoire LOCEAN/IPSL, Sorbonne Université-CNRS-IRD-MNHN, Paris, 75005, France
- 2 MARBEC, Université de Montpellier, CNRS, Ifremer, IRD, 34203 Sète, France
- 3 Laboratoire LSCE/IPSL, CEA-CNRS-UVSQ, Université Paris-Saclay Gif-sur-Yvette, 91191, France

Correspondence to: Nicolas Metzl (nicolas.metzl@locean.ipsl.fr)

## **Abstract:**

New observations obtained in 2021 and 2022 are presented and used to investigate the trend of the carbonate system (including pH<sub>T</sub> and aragonite saturation state,  $\Omega_{ar}$ ) in the southern sector of the Mozambique Channel. Using historical and new data in April-May we observed an acceleration of the acidification ranging from -0.012 decade<sup>-1</sup> in 1963-1995 to -0.027 (±0.003) decade<sup>-1</sup> in 1995-2022. Result from a neural network (FFNN) model for all seasons also suggests faster pH trend in recent decades, -0.011 decade<sup>-1</sup> over 1985-1995 and -0.018 decade<sup>-1</sup> over 1995-2022. In May 2022 we estimated  $\Omega_{ar}$  of 3.49, about 0.3 lower than observed in May 1963 ( $\Omega_{ar}$  = 3.86). The lowest  $\Omega_{ar}$  value of 3.23 was evaluated from the FFNN model in September 2023 that corresponds to the hypothetical critical threshold value (3.25) for coral reefs. In 2025 a marine heat wave was observed in this region (sea surface temperature up to 30°C) and data from a BGC-Argo float indicate that sea surface pH was the lowest in January 2025 (pH<sub>T</sub> = 7.95) whereas  $\Omega_{ar}$  was the lowest in Mach 2025 ( $\Omega_{ar}$  = 3.2). A projection of the C<sub>T</sub> concentrations based on observed anthropogenic CO<sub>2</sub> in subsurface water and future anthropogenic CO<sub>2</sub> emissions scenario, suggests that a risky level for corals ( $\Omega_{ar}$  < 3) could be reached as soon as year 2034.

**<u>Keywords:</u>** Ocean Acidification, Decadal Trend, Mozambique Channel

#### 1 Introduction:

The ocean plays a major role in reducing the impact of climate change by absorbing more than 90% of the excess heat in the climate system (Cheng et al., 2025; Forster et al 2025) and about 25% of human released  $CO_2$  (Friedlingstein et al., 2025). The oceanic  $CO_2$  uptake also changes the chemistry of seawater reducing its buffering capacity (Revelle and Suess, 1957) and leading to a process known as ocean acidification (OA) with potential impacts on marine organisms and ecosystems (Fabry et al., 2008; Doney et al., 2009, 2020; Gattuso et al, 2015; Schönberg et al, 2017; Cornwall et al, 2021). Global ocean models or Earth System Models predict that, due to future anthropogenic  $CO_2$  emissions and global warming, the sea surface pH could decrease by 0.4 and aragonite saturation state ( $\Omega$ ar) could be as low as 3 in the tropics by 2100 (Hoegh-Guldberg et al, 2007; Kwiatkowski et al, 2020; Jiang et al, 2023; Findlay et al, 2025). However, current global ocean models cannot fully replicate observations and not yet simulating all processes that govern ocean acidification (e.g. seasonal cycles of  $C_T$  and  $A_T$ , accumulation of  $C_{ant}$ , etc...). Long-term observations of the carbonate system are needed to compare and validate model results (Tilbrook et al, 2019).

The first estimate of the decadal pH change based on  $CO_2$  fugacity (fCO<sub>2</sub>) observations in the global ocean (using SOCAT data, Bakker et al, 2014) suggests a decrease of pH ranging between - 0.003 decade<sup>-1</sup> (±0.005) in the North Pacific and -0.024 decade<sup>-1</sup> (±0.005) in the Indian Ocean over

1981-2011 (Lauvset et al, 2015). Reconstruction methods also based on SOCAT observations evaluated a global ocean decrease of pH in surface waters of -0.0181 ( $\pm$  0.0001) decade<sup>-1</sup> (Iida et al, 2021), -0.0166 ( $\pm$  0.0010) decade<sup>-1</sup> (Ma et al, 2023) and -0.017 ( $\pm$  0.004) decade<sup>-1</sup> (Chau et al, 2024). These studies also highlighted the regional differences of the pH and aragonite saturation state ( $\Omega$ ar) trends. This calls for dedicated studies at regional scale in order to better interpret the inter-annual to multi-decadal changes of the oceanic carbonate system as the trends and associated uncertainties depend on the data available. Compared to other basins, observations are sparse in the Indian Ocean (Lauvset et al, 2015; Bakker et al, 2016, 2024). However, thanks to a new cruise conducted in 2019, it has been shown that the Mozambique Channel experienced an acceleration with respect to  $\Theta$  the acidification in recent years, a pH trend of -0.023 decade<sup>-1</sup> ( $\pm$ 0.005) over 1995–2019 (Lo Monaco et al, 2021). In a more recent analysis Chakraborty et al (2024) used several methods, including a high resolution model dedicated to the Indian Ocean and found an acceleration of the pH trend of -0.011 ( $\pm$ 0.00) decade<sup>-1</sup> in 1980–1989 to -0.019 ( $\pm$ 0.004) decade<sup>-1</sup> in 2010–2019. Both studies concluded that strengthening of acidification trend was mainly driven by ocean CO<sub>2</sub> uptake.

In this study, we present new data obtained in January 2021 and April-May 2022 in the Mozambique Channel and used the results of a FFNN model (Chau et al, 2024) extended to 2023 to explore the decadal trends of the carbonate system over 1963-2023. We also use these data to validate a projection of the acidification in the near future. To highlight  $CO_2$  source anomalies when the ocean was exceptionally warm, results from a BGC-Argo float in the Mozambique Channel in 2024-2025 are also presented.

#### 2 Data selection and methods

#### 2.1 Data selection

To explore the long-term change of the carbonate system in this region, we selected the fCO $_2$  SOCAT data, version v2024 (Bakker et al, 2016, 2024). With recent cruises conducted on-board the ship Marion-Dufresne in January 2021 (OISO-31) and April-May 2022 (RESILIENCE) this includes 10 cruises in the Mozambique Channel (Table 1 and Figure 1). Some of these cruises were previously described to analyze the distribution air-sea  $CO_2$  fluxes and pH changes in the Mozambique Basin and the African coastal zone (Metzl et al, 2025b). Here we focus on the data obtained in the southern Mozambique Channel. To complete the shipboard data after 2022 we also used data from a BGC-Argo float (WMO ID 7902123) that was launched onboard R/V Sonne in the Mozambique Channel in late 2024. During some cruises (2004, 2019 and 2021) continuous underway  $A_T$  and  $C_T$  measurements were also performed (data available in Metzl et al, 2025a). These  $A_T$  and  $C_T$  data are used to compare and validate results of the pH trends based on fCO $_2$  data.

Table 1: List of cruises in the Mozambique Channel from SOCAT-v2024 (Bakker et al, 2024).

| 93  |                  |       |        |                                     |
|-----|------------------|-------|--------|-------------------------------------|
| 94  | EXPOCODE         | Month | n Year | Reference or Principal Investigator |
| 95  |                  |       |        |                                     |
| 96  | 31AR19630216     | 5     | 1963   | Keeling and Waterman (1968)         |
| 97  | 316N19950611     | 6     | 1995   | Key, R.                             |
| 98  | 33RO19990211     | 2     | 1999   | Wanninkhof, R.                      |
| 99  | 49NZ20031209     | 12    | 2003   | Murata, A.                          |
| 100 | 35MF20040106(a)  | 1     | 2004   | Metzl (2009)                        |
| 101 | 06BE20140710     | 7     | 2014   | Steinhoff, T., Koertzinger, A       |
| 102 | 33RO20180423     | 4     | 2018   | Wanninkhof, R., Pierrot, D.         |
| 103 | 35MV20190405 (a) | 4     | 2019   | Lo Monaco et al. (2021)             |
| 104 | 35MV20210113 (a) | 1     | 2021   | Metzl et al. (2025b)                |
| 105 | 35MV20220420     | 4     | 2022   | Metzl et al. (2025b)                |
| 106 |                  |       |        |                                     |

(a) For these cruises underway A<sub>T</sub> C<sub>T</sub> data available at https://doi.org/10.17882/102337

Time [yr]

Figure 1: Left: Tracks of cruises in the Mozambique Channel in the SOCAT data-base, version v2024 (Bakker et al., 2016; 2024). This includes recent OISO-31 and RESILIENCE cruises in 2021 and 2022. Color code is for Year. Black circles identified the coral reefs locations. Right: Tracks of cruises near the coral reefs area. Figures produced with ODV (Schlitzer, 2018).

35°E

40°E

2.2 Methods

The methods for surface underway  $fCO_2$  and  $A_T$   $C_T$  measurements were described in previous studies (e.g. Lo Monaco et al, 2021). For  $fCO_2$  measurements during OISO-11 (2004), CLIM-EPARSES (2019), OISO-31 (2021) and RESILIENCE (2022) cruises, sea-surface water was continuously equilibrated with a "thin film" type equilibrator thermostated with surface seawater (Poisson et al., 1993) and xCO2 in the dried gas was measured with a non-dispersive infrared analyzer (NDIR, Siemens Ultramat 6F). Standard gases for calibration (around 280 ppm, 350 ppm and 490 ppm) were

measured every 6 hours. The sea surface temperature (SST) and equilibrium temperature were measured using SBE21 and SBE38 probes (accuracy  $0.002^{\circ}$ C) respectively. During the RESILIENCE cruise the difference of SST and equilibrium temperature was on average +0.088 ±0.066 °C (n= 6416). For all cruises, the sea surface salinity (measured with SBE21) was regularly checked with discrete samples and has been corrected if some drift was observed. The fCO<sub>2</sub> in situ data were corrected for warming using corrections proposed by Copin-Montégut (1988, 1989). Note that when incorporated in the SOCAT data-base, the original fCO2 data are recomputed (Pfeil et al., 2013) using temperature correction from Takahashi et al. (1993). Given the very small difference between equilibrium temperature and sea surface temperature, the fCO<sub>2</sub> data from SOCAT used in this analysis (Bakker et al., 2024) are almost identical (within 1  $\mu$ atm) to the original fCO<sub>2</sub> values.

During 3 cruises, in January 2004 (OISO-11), April 2019 (CLIM-EPARSES) and January 2021 (OISO-31),  $A_T$  and  $C_T$  were measured continuously in surface water using a potentiometric titration method (Edmond, 1970) in a closed cell. For calibration, we used the Certified Referenced Materials provided by Pr. A. Dickson (SIO, University of California). Based on repeatability from duplicate analyses of continuous sea surface sampling at the same location (when the ship was stopped) we estimated the accuracy for both  $A_T$  and  $C_T$  better than 4 µmol.kg<sup>-1</sup> (Metzl et al, 2025a). The  $A_T$  and  $C_T$  data for these cruises are available at the Seanoe platform (https://doi.org/10.17882/102337). These data offered comparisons and validation for the calculations of the carbonate system properties using fCO<sub>2</sub> data and  $A_T$ /Salinity relationship.

# 2.3 Carbonate system calculation and A<sub>T</sub>/Salinity relationship

When two of the carbonate system properties are measured (here either fCO<sub>2</sub>,  $A_T$  or  $C_T$ ) they can be used to calculate other species and the saturation state of aragonite ( $\Omega_{ar}$ ). Here we used the CO2sys program (version CO2sys\_v2.5, Orr et al., 2018) with K1 and K2 dissociation constants from Lueker et al. (2000) and KSO4 constant from Dickson (1990). The total boron concentration is calculated according to Uppström (1974). When using fCO<sub>2</sub> data to derive pH<sub>T</sub> (pH<sub>T</sub> for Total Scale) or C<sub>T</sub>, one needs A<sub>T</sub> concentrations that can be derived from salinity (e.g. Millero et al, 1998). Here we used the A<sub>T</sub>/Salinity relationship adapted to the Mozambique Channel (Lo Monaco et al, 2021).

 $A_T$  (µmol.kg-1) = 73.841 (± 1.15) \* SSS – 291.02 (± 40.4) (n= 548,  $r^2$  = 0.88) (Eq. 1)

# 2.4. CMEMS-LSCE-FFNN model

The fCO $_2$  data are not available each year and only for few seasons (Table 1). To complete the observations we used the results from an ensemble of feed-forward neural network model (CMEMS-LSCE-FFNN or FFNN for simplicity here, Chau et al., 2024). Based on the SOCAT gridded datasets this model composes surface ocean carbonate system fields at 0.25 x 0.25 square degree resolution and monthly scale. The reconstructed fCO $_2$  is used to derive monthly surface C $_T$ , pH $_T$  and aragonite and calcite saturation states, as well as air-sea CO $_2$  fluxes. A full description of the model is presented in Chau et al (2024) and the datasets including uncertainties are available under the DOI https://doi.org/10.48670/moi-00047.

#### 3 Results and discussion

## 3.1 A Repeated line in 2019 and 2022

In April 2019 and 2022 underway measurements were conducted for  $fCO_2$ . The measurements of  $A_T$  and  $C_T$  were also performed in 2019. The tracks of the cruises enabled to select the data obtained along the same track and for the same season in the southern Channel in order to compare the observations three years apart (Figure 2). Given the variability observed around Europa

Island and the front identified at 22.5°S in April 2019 (Figure 2) the data were averaged in the band 23°-26°S. The mean values over the same latitudinal band (23-26°S) show significant differences between 2019 and 2022 (Table 2). In 2022 the ocean was slightly colder and saltier. Consequently,  $A_T$  concentrations were also higher in 2022 but the salinity normalized  $A_T$  (N- $A_T$  normalized at salinity 35) were the same with a difference of +1.1  $\mu$ mol.kg<sup>-1</sup>. As expected, due to the  $CO_2$  uptake, the oceanic  $fCO_2$  and  $C_T$  concentrations were higher in 2022 and  $pH_T$  was lower. The increase of oceanic  $fCO_2$  over 3 years (+7.9  $\mu$ atm) was almost the same as in the atmosphere (+7.0  $\mu$ atm). At constant  $A_T$ , salinity and temperature, the observed  $fCO_2$  change would translate in an increase of +4.4  $\mu$ mol.kg<sup>-1</sup> for  $C_T$  when we observed an increase of +18  $\mu$ mol.kg<sup>-1</sup> (Table 2). We interpret this difference as being due to the regional circulation. In April 2019 southward currents would import low  $C_T$  and  $A_T$  whereas in April 2022 northward currents would transport colder and saltier waters with higher  $C_T$  and  $A_T$ . This reversed circulation is confirmed with the ADCP data recorded during the cruises as well as from the geostrophic currents (Metzl et al, 2022; Ternon et al, 2023).

For pH<sub>T</sub>, the decrease of -0.005 over three years, i.e. -0.0017 yr<sup>-1</sup>, is surprisingly close to what is generally observed at global scale and over several decades (-0.0017  $\pm 0.0004$ .yr<sup>-1</sup>, Chau et al, 2024). Finally, we note that the difference between 2019 and 2022 measurements is much higher than that obtained when comparing measured and calculated A<sub>T</sub> C<sub>T</sub> values (Table 2). This confirms the use of fCO<sub>2</sub> data and adapted A<sub>T</sub>/S relationship to derive the carbonate system properties in this region (Lo Monaco et al, 2021; Metzl et al, 2025b), and to explore the seasonal cycles and long-term trends described in the next sections.

\_\_\_\_\_\_

Table 2: Mean values of underway sea surface observations and their difference obtained along the same track in 2019 and 2022 in the region 23-26°S (see Figure 2). Nb is the number of data. Standard-deviations are in bracket. For 2019 (CLIM-EPARSES cruise), the results from underway  $A_T-C_T$  measurements are listed allowing calculation of  $fCO_2$  and pH based on the  $A_T-C_T$  pairs which permit comparisons with those derived from  $fCO_2$  measurements.

| Cruise Period                               | Nb  | SST     | SSS     | $A_{T}$ | $C_T$           | $fCO_2$ | $pH_T$  | Atm. xCO2 |
|---------------------------------------------|-----|---------|---------|---------|-----------------|---------|---------|-----------|
|                                             |     | °C      | -       | μmol.kį | g <sup>-1</sup> | μatm    | TS      | ppm       |
|                                             |     |         |         |         |                 |         |         |           |
| RESILIENCE fCO <sub>2</sub>                 | 282 | 26.765  | 35.423  | 2324.6  | 2002.0          | 394.8   | 8.047   | 414.7     |
| April 2022                                  |     | (0.608) | (0.048) | (3.6)   | (5.0)           | (3.4)   | (0.003) |           |
| CLIM-EPARSES fCO <sub>2</sub>               | 294 | 27.497  | 35.288  | 2314.7  | 1984.0          | 386.9   | 8.053   | 407.5     |
| April 2019                                  |     | (0.341) | (0.084) | (6.2)   | (7.5)           | (4.9)   | (0.004) |           |
| CLIM-EPARSES A <sub>T</sub> -C <sub>T</sub> | 70  | 27 //01 | 25 272  | 2314.2  | 1086 1          | 380 7   | 8.050   |           |
| April 2019                                  | 70  |         |         | (6.7)   |                 | (7.4)   | (0.006) |           |
| Difference Method 202                       | 10  | +0 006  | ±0.016  | +0.5    | -2.1            | -2.9    | +0.002  |           |
| Difference Method 20.                       | 19  | +0.096  | +0.016  | +0.5    | -2.1            | -2.9    | +0.002  |           |
| Difference 2022-2019                        |     | -0.732  | +0.135  | +10.0   | +18.0           | +7.9    | -0.005  | +7.2      |

Figure 2: Distribution of measured fCO<sub>2</sub> ( $\mu$ atm), calculated  $C_{T}$  ( $\mu$ mol/kg) and calculated pH<sub>T</sub> (TS) along a repeated track in April 2019 (blue) and April 2022 (red) in the southern Mozambique Channel. The dashed red line is for atmospheric fCO<sub>2</sub> in 2022. The underway  $C_{T}$  measurements in 2019 are also shown (open circles) as well as pH<sub>T</sub> calculated using measured A<sub>T</sub> and  $C_{T}$ . Average values for the latitudinal band 23-26°S are presented in Table 2.

## 3.2 Seasonal variations

In the Mozambique Channel, where SST presents large seasonal variations (up to 4°C), fCO<sub>2</sub> is mainly controlled by temperature like in the Indian subtropics (e.g. Metzl et al 1998; Takahashi et al, 2002; Bates et al, 2006). In this region, observations are not available for all seasons (Table 1) but the seasonal range derived from the climatology (Fay et al, 2024) or the FFNN model (Chau et al, 2024) is coherent compared to the data (Figure 3, Figure S1). The observations and the models indicate that between January and July fCO<sub>2</sub> decreases by about 50  $\mu$ atm while pH<sub>T</sub> increases (0.03 to 0.04 units). This is a large signal compared to the expected decadal change (about +20  $\mu$ atm.decade<sup>-1</sup> for fCO<sub>2</sub> and -0.017.decade<sup>-1</sup> for pH<sub>T</sub>); therefore, to derive and interpret the trends, data have to be selected for the same season. As opposed to fCO<sub>2</sub>, C<sub>T</sub> presents lower concentrations in February-April and higher ones in July-August (Figure 4). When the mixed-layer depth (MLD) is shallow in December-March the decrease of C<sub>T</sub> is probably linked to biological activity but this is not clearly quantified (Lo

Monaco et al, 2021). The progressive  $C_T$  increase of about +30  $\mu$ mol.kg<sup>-1</sup> from March to August is likely driven by vertical mixing when MLD is deeper in austral winter (Figure 4).

This seasonality was well observed from repeated measurements at stations located along 25°S in June 1995 and December 2003 (Figure S2). In June 1995 when the MLD reached 80 m,  $C_T$  concentrations were homogeneous within the MLD layer. The same was true for the anthropogenic  $CO_2$  concentrations ( $C_{ant}$ ) here evaluated using the TrOCA method (Touratier et al. 2007). On the opposite, in December 2003, when the MLD was shallower,  $C_T$  presented a sharp increase within the subsurface layer whereas  $C_{ant}$  concentrations were unrealistic in surface seawaters. Although from 1995 to 2003 the  $C_T$  concentrations would increase by around +7  $\mu$ mol/kg due to the anthropogenic  $CO_2$  uptake in that region (Murata et al, 2020; Metzl et al, 2025b), N- $C_T$  (normalized  $C_T$  at salinity 35) in June 1995 were almost the same as in December 2003 coherent with the seasonal cycle derived from the climatology (Figure 4).

The seasonal variations of  $fCO_2$  and  $pH_T$  in the Mozambique Channel appear thus linked to both temperature and mixing process with competition between the two drivers (Figure S3). In addition to the rising atmospheric  $CO_2$ , these two processes probably also drive inter-annual, decadal and long-term change of  $fCO_2$  and  $pH_T$  in the region as the Indian Ocean experienced a pronounced warming (Cheng et al., 2025). Specifically, in the southern Mozambique Channel the SST has increased by  $+0.11 \pm 0.009$  °C per decade since the 1960s (Figure S4), a signal that should be taken into account when interpreting the decadal trends of carbonate properties and  $CO_2$  fluxes. In January 2025 the SST anomaly reached +1.6°C at 25°S in the Channel.

Figure 3: Seasonal cycle of (a)  $fCO_2$  ( $\mu$ atm) and (b)  $pH_T$  in the southern Mozambique Channel (24-30°S). Average observations are presented for each cruise (colored circles). The full seasonal cycles are shown for the monthly climatology (reference year 2010, Fay et al, 2024) and for the FFNN model for years 2010 and 2022 with respective error bars.

Figure 4: Seasonal cycle of  $C_T$  (µmol.kg<sup>-1</sup>) in the southern Mozambique Channel (24-30°S). Average observations are presented for each cruise (colored circles). The full seasonal cycles are shown based on the monthly climatology for a reference year 2010 (Fay et al, 2024) and the FFNN-LSCE model for year 2010 (Chau et al, 2024). The mixed-layer depth (MLD in m, blue line) is averaged in this region (from multi-year reprocessed monthly data, ARMOR3D L4, https://doi.org/10.48670/moi-00052, last access 20/4/2025).

## 3.3 Trends in the Southern Mozambique Channel (1963-2023)

In this region, the ocean is a permanent  $CO_2$  sink leading to a gradual increase of  $C_T$  concentrations and decrease of  $pH_T$ . The air-sea  $CO_2$  flux derived from the FFNN model is on average -0.249 (±0.063) molC.m<sup>-2</sup>.yr<sup>-1</sup> (Figure 5) in the range of the climatology (-0.3 molC.m<sup>-2</sup>.yr<sup>-1</sup>, Fay et al, 2024). The FFNN model also suggests that the sink reinforced over 2016-2021 with a perceptible faster increase of  $C_T$  (Figure S5).

Figure 5: Time-series of air-sea  $CO_2$  flux (black, negative for ocean sink) and  $C_T$  concentration (red) averaged in the southern Mozambique Channel (24-30°S) based on the FFNN-LSCE model over 1985-2023. For  $C_T$ , the result is presented with a 36-month running mean. Also shown is the climatological value of the flux for year 2010 in this region (red circle, Fay et al, 2024).

# 3.3.1 Decadal trend from fCO<sub>2</sub> and A<sub>T</sub> C<sub>T</sub> data: January 2004 and 2021

We started the analysis of the decadal change by comparing observations obtained in January 2004 and 2021 when data were available for both underway fCO<sub>2</sub>,  $A_T$  and  $C_T$  measurements. The comparison is focused along the tracks occupied in the same region (27-29°S/40-43°E, Figure 6,

Table 3). For both cruises the differences between measurements and calculations are in the range of the errors in the CO2sys calculations (errors on measurements and constants K<sub>1</sub>, K<sub>2</sub>, Orr et al., 2018). For example, in January 2004 the pH<sub>T</sub> calculated with A<sub>T</sub> and C<sub>T</sub> measurements was 8.069 against 8.064 when using the fCO<sub>2</sub> data and the A<sub>T</sub>/S relationship. In 2021 the pH<sub>T</sub> were respectively 8.030 and 8.032 (Table 3). For C<sub>T</sub> the difference between calculated and measured C<sub>T</sub> was only 4.4 μmol.kg <sup>1</sup> in 2004 and -1.9 μmol.kg<sup>-1</sup> in 2021 when the observed increase over 17 years is around 28 μmol.kg<sup>-1</sup> <sup>1</sup>. We noticed that in 2021, the properties present a high variability along the track linked to the presence of eddies. The  $C_T$  and  $A_T$  concentrations could vary by about 20 to 40  $\mu$ mol.kg<sup>-1</sup> at mesoscale but this has a small impact on calculated fCO<sub>2</sub> and pH<sub>T</sub>, and when properties are averaged along the track (Table 3). For both periods the ocean fCO2 was close to atmospheric CO2, i.e. near equilibrium (fCO<sub>2</sub> ocean-fCO<sub>2</sub> atm =  $\Delta$ fCO<sub>2</sub> = -0.04 ±3.11 µatm in 2004 and 0.37 ±10.04 µatm in 2021). Although there were some differences of pH<sub>T</sub> calculated from the two data-sets (underway fCO<sub>2</sub> or A<sub>T</sub>  $C_T$  data), the estimated  $pH_T$  change of -0.032 or -0.040 over 17 years was large compared to the uncertainty of the CO2sys calculation. This would correspond to a pH<sub>T</sub> trend varying between -0.0019 and -0.0023.yr<sup>-1</sup>. This comparison of observations in January 2004 and 2021 supports the use of the selected  $A_T/S$  relationship for pH calculations based on all  $fCO_2$  data available over 1963–2023 in order to explore the long-term trend described in the next section.

 Table 3: Mean values of underway sea surface observations and their difference obtained in the same region (27-29°S/40-43°E) in January 2004 and 2021. Nb is the number of data. Standard-deviations are in bracket. The results are presented for both methods (underway fCO<sub>2</sub> or  $A_T$ - $C_T$  measurements) and fCO<sub>2</sub>, pH<sub>T</sub> calculated with  $A_T$ - $C_T$  pairs compared with those derived from fCO<sub>2</sub> measurements. The last lines are the difference for 2021 minus 2004 and errors associated to the measurements or calculations (\*).

| Cruise Method                                                           | Nb  | SST          | SSS          | A <sub>T</sub> | C <sub>T</sub>  | fCO <sub>2</sub> | pH <sub>⊤</sub>  | Atm. xCO <sub>2</sub> |
|-------------------------------------------------------------------------|-----|--------------|--------------|----------------|-----------------|------------------|------------------|-----------------------|
| Period                                                                  |     | °C           | -            | μmol.k         | g <sup>-1</sup> | μatm             | TS               | ppm                   |
| OISO-11 fCO <sub>2</sub>                                                | 140 | 27 202       | 35.282       | 2214.2         | 1079 E          | 274.7            | 8.064            | 374.8                 |
| January 2004                                                            | 140 |              | (0.050)      |                | (6.7)           | (3.2)            | (0.002)          |                       |
| January 2004                                                            |     | (0.551)      | (0.030)      | (3.7)          | (0.7)           | (3.2)            | (0.002)          |                       |
| OISO-11 A <sub>T</sub> -C <sub>T</sub>                                  | 30  | 27.516       | 35.248       | 2315.4         | 1974.1          | 368.7            | 8.069            |                       |
| January 2004                                                            |     | (0.609)      | (0.042)      | (7.2)          | (9.8)           | (7.6)            | (0.007)          |                       |
| •                                                                       |     | ,            | . ,          | . ,            | . ,             |                  | , ,              |                       |
| OISO-31 fCO <sub>2</sub>                                                | 102 | 27.825       | 35.508       | 2330.9         | 2006.8          | 412.5            | 8.032            | 412.2                 |
| January 2021                                                            |     | (0.793)      | (0.118)      | (8.7)          | (14.1)          | (10.0)           | (0.008)          |                       |
|                                                                         |     |              |              |                |                 |                  |                  |                       |
| OISO-31 A <sub>T</sub> -C <sub>T</sub>                                  | 17  |              | 35.515       |                |                 |                  | 8.030            |                       |
| January 2021                                                            |     | (0.678)      | (0.139)      | (9.6)          | (18.7)          | (21.2)           | (0.017)          |                       |
| D:ff 2004 2004                                                          |     |              |              |                |                 |                  |                  |                       |
| Difference 2021-2004                                                    |     | 0.500        | 0.225        | 467            | 20.2            | 27.0             | 0.000            | 27.4                  |
| Underway fCO <sub>2</sub>                                               |     | 0.532        | 0.225        | 16.7           | 28.3            | 37.8             | -0.032           | 37.4                  |
| Underway A <sub>T</sub> -C <sub>T</sub>                                 |     | 0.400        | 0.267        | 11.5           | 29.4            | 45.7             | -0.040           |                       |
| Error using fCO                                                         |     | 0.01         | 0.01         | 4              | 72*             | 2                | 0.014*           |                       |
| Error using fCO <sub>2</sub> Error using A <sub>T</sub> -C <sub>T</sub> |     | 0.01<br>0.01 | 0.01<br>0.01 | 4              | 7.3 *<br>4      | 2<br>13.9*       | 0.014*<br>0.007* |                       |
| LITOI USING AT-CT                                                       |     | 0.01         | 0.01         | 4              | 4               | 13.5             | 0.007            |                       |
|                                                                         |     |              |              |                |                 |                  |                  |                       |

Figure 6: Distribution of measured or calculated  $C_T$  (a,  $\mu$ mol  $kg^{-1}$ ),  $A_T$  (b,  $\mu$ mol  $kg^{-1}$ ),  $fCO_2$  (c,  $\mu$ atm) and  $pH_T$  (d) along the same track in January 2004 (black symbols) and January 2021 (grey symbols). Values derived from  $fCO_2$  measurements are in filled symbols/lines, those from the  $A_T$   $C_T$  measurements in open symbols/dashed lines. In (c) the red lines represent the atmospheric  $CO_2$  in 2004 and 2021. Average values and their differences are presented in Table 3.

# 3.3.2 Multi-decadal trends from fCO<sub>2</sub> data (1963-2023)

 For long-term trends, we used fCO<sub>2</sub> observations and observations-based reconstructions averaged in the region 24-30°S (Table 4). As the observations are not available for all seasons, we selected the period April-May to calculate the trends from the data (same season for the first and the last cruises in 1963 and 2022) whereas the FFNN model offers information for all seasons. Back in the 1960s, the observations in 1963 indicate that the ocean was a CO<sub>2</sub> sink in May (Figure 7a), the value of  $\Delta$ fCO<sub>2</sub> = -32.2  $\mu$ atm being almost the same as observed in May 2022 ( $\Delta$ fCO<sub>2</sub> = -32.5  $\mu$ am). This suggests a strong link between ocean and atmospheric fCO<sub>2</sub> (Figure S6).

For the first observational period, the changes between 1963 and 1995 indicated a pH<sub>T</sub> decrease of -0.040. Over 32 years this pH<sub>T</sub> change was driven by the C<sub>T</sub> increase (effect on pH<sub>T</sub>= -0.045), the A<sub>T</sub> increase (effect on pH<sub>T</sub>= +0.012) and the warming of 0.95°C (effect on pH<sub>T</sub>= -0.015). Between 1995 and 2022 the observed decrease accelerated to -0.0027 (±0.0003) yr<sup>-1</sup> (Table 4). In contrast, the neural network suggested smaller pH<sub>T</sub> trends. However, as in the observations, the annual pH<sub>T</sub> change from the model was faster in recent decades (-0.0018 yr<sup>-1</sup> over 1995-2022 against -0.0011 yr<sup>-1</sup> over 1985-1995, Table 4). The model also suggested different trends depending on the season. The pH<sub>T</sub> trend appeared indeed faster in July (when the ocean CO<sub>2</sub> sink is stronger) than in January or April (Table 4).

The new data in 2021 and 2022 and the FFNN model extended to 2023 confirmed a previous analysis in the Mozambique Channel (Lo Monaco et al, 2021) with a pH $_{\rm T}$  trend of -0.0023 yr $^{-1}$  (±0.00048) over 1995–2019. Our new results in the southern Mozambique Channel are also in the range of the pH $_{\rm T}$  trends previously evaluated at basin scale in the Indian Ocean, -0.0027 yr $^{-1}$  (±0.0005) over 1991-2011 (Lauvset et al, 2015). High resolution ocean models applied to the northern Indian Ocean also suggest an acceleration of the acidification, with pH $_{\rm T}$  trend reaching -0.0019 yr $^{-1}$  (±0.0004) in 2010–2019 (Chakraborty et al, 2024), somehow lower than our estimate based on observations at regional scale in the Mozambique Channel (-0.0027 yr $^{-1}$  ±0.0003 in 1995-2022, Table 4).

Table 4: Trends of properties in the southern Mozambique Channel derived from observations and the FFNN model. For observations, the trends are evaluated for April-May season only. For FFNN, trends are estimated for all seasons or only for January, April, May and July. Standard-deviations are in bracket.

| Method        | Period    | fCO <sub>2</sub><br>μatm.yr <sup>-1</sup> | C <sub>T</sub><br>μmol.k | A <sub>T</sub><br>g.yr <sup>-1</sup> | pH <sub>T</sub>     |
|---------------|-----------|-------------------------------------------|--------------------------|--------------------------------------|---------------------|
| Obs April-May | 1963-1995 | 1.11                                      | 0.91                     | 0.52                                 | -0.0012             |
| Obs April-May | 1963-2022 | 1.84<br>(0.21)                            | 0.69<br>(0.20)           | 0.08<br>(0.13)                       | -0.0020<br>(0.0002) |
| Obs April-May | 1995-2022 | 2.57<br>(0.30)                            | 0.49<br>(0.52)           | -0.34<br>(0.22)                      | -0.0027<br>(0.0003) |
| FFNN annual   | 1985-2023 | 1.76<br>(0.05)                            | 0.99<br>(0.04)           | 0.02<br>(0.02)                       | -0.0017<br>(0.0001) |
| FFNN annual   | 1985-1995 | 1.15<br>(0.34)                            | 1.03<br>(0.29)           | 0.00 (0.08)                          | -0.0011<br>(0.0004) |
| FFNN annual   | 1995-2022 | 1.84<br>(0.09)                            | 1.10<br>(0.07)           | 0.06<br>(0.03)                       | -0.0018<br>(0.0001) |
| FFNN January  | 1985-2023 | 1.61<br>(0.03)                            | 0.75<br>(0.07)           | 0.00<br>(0.05)                       | -0.0015<br>(0.0000) |
| FFNN April    | 1985-2023 | 1.74<br>(0.03)                            | 1.01<br>(0.07)           | 0.03<br>(0.07)                       | -0.0017<br>(0.0000) |
| FFNN May      | 1985-2023 | 1.71<br>(0.03)                            | 1.07                     | 0.07 (0.05)                          | -0.0017<br>(0.0000) |
| FFNN July     | 1985-2023 | 1.97<br>(0.04)                            | 1.17<br>(0.05)           | 0.04 (0.02)                          | -0.0020<br>(0.0000) |

The aragonite saturation state  $(\Omega_{ar})$  was lower during austral summer (July-September). In May 1963, we estimated an aragonite saturation state of 3.86 (Figure 7c). It dropped to 3.49 in May 2022, a value close to that observed in July 2014 (3.47). The lowest  $\Omega_{ar}$  value of 3.23 was identified in September 2023 from the FFNN model. At that period,  $\Omega_{ar}$  was lower than 3.3 in the south of 20°S in the Mozambique Channel (Figure S7). This is close to the hypothetical critical threshold of  $\Omega_{ar}$  = 3.25, i.e. a risky level for coral reefs in the ocean claimed by Hoegh-Guldberg et al., (2007). Note that there are reefs know to thrive at  $\Omega_{ar}$  <3.0 like at volcanic  $CO_2$  seeps in Papua New Guinea ( $\Omega_{ar}$  = 2.41, Strahl et al 2015; see also review by Camp et al. 2018) but that their species composition and coral cover are different than at ambient conditions (i.e.  $\Omega_{ar}$ >3.3 considering Hoegh-Gulberg et al 2007). However, Strahl et al (2015) showed that calcification rate seems to vary among coral species, suggesting take conclusions of Hoegh-Gulberg et al (2007) with caution. With an annual trend of 0.010.yr $^{-1}$  for  $\Omega_{ar}$  over 1985-2023, a value of 3.3 would be reached in 2060 in summer whereas it was already observed in 2020 in winter with possible consequences on reef species composition and functioning (Tribollet al 2009, 2019; Schönberg et al 2017; Camp et al. 2018; Eyre et al. 2018; Cornwall et al 2021).

Although there are differences depending on the season and the method (in-situ observations, extrapolation of sparse in-situ observations through a FFNN model) all results suggest

an acceleration of the acidification in the last few years (Table 4, Figure 7) and a decrease of  $\Omega_{ar}$  that are mainly driven by the  $C_T$  increase through continuous ocean  $CO_2$  uptake (Ma et al, 2023). Given the rapid change of atmospheric  $CO_2$  in the recent years (up to +3.77 ppm.yr<sup>-1</sup> in 2024, Lan et al, 2025) how the carbonate system will change in the near future in this region and will impact corals reefs that are abundant (from Europa to Mayotte in the Mozambique Chanel) and subject to global warming, marine heat waves (e.g. Mawren et al, 2022; Alaguarda et al. 2022), ocean acidification, higher frequency of tropical cyclones and anthropogenic pressures (overfishing for instance), remains an important question.

Figure 7: Time-series of fCO<sub>2</sub> (a), pH<sub>T</sub> (b) and  $\Omega_{ar}$  (c) in the southern Mozambique Channel (24-30°S) based on averaged observations (circles) and the FFNN-LSCE model over 1985-2023. In (a) the red line represents the atmospheric CO<sub>2</sub>. Available observations are shown for all seasons but the trends (Table 4) evaluated using only April-May data (red circles).

#### 3.4 Projection in the near future

A recent analysis found that the  $C_{ant}$  concentrations in subsurface water in the Mozambique Basin were positively related to atmospheric  $CO_2$  with a slope of +0.512 ±0.050 µmol kg<sup>-1</sup> µatm<sup>-1</sup> (Metzl et al, 2025b, Figure S8). Here, we assume that this relationship is valid for the southern Mozambique Channel. To reconstruct the past and future change of the carbonate system properties we calculated the  $C_T$  concentration over 1960-2100 by correcting  $C_T$  for each year using the relationship between  $C_{ant}$  and atmospheric  $CO_2$ .

$$C_T(t) = C_T(t-1) + C_{ant}(t) - C_{ant}(t-1)$$
 (Eq. 2)

For future atmospheric  $CO_2$ , we used two SSP emissions scenarios (Shared Socioeconomic Pathways, Meinshausen et al., 2020), a "high" emission scenario SSP5-8.5 and a stabilization scenario SSP2-4.5 (Figure 8a). To explore the change of the aragonite saturation state, we applied this model (Eq. 2) for August when  $\Omega_{ar}$  is the lowest. Temperature and salinity were fixed from the climatology in August (SST =22.685 °C; SSS = 35.303) and fCO<sub>2</sub>, pH<sub>T</sub> and  $\Omega_{ar}$  were calculated each year with the C<sub>T</sub> A<sub>T</sub> pairs using version CO2sys\_v2.5 (Orr et al, 2018). The reconstructed C<sub>T</sub>, fCO<sub>2</sub>, pH<sub>T</sub> and  $\Omega_{ar}$  for August compared well with the observations (in July) and with the FFNN model in August (Figure 8; Table S1, Figure S9) indicating that the simulation captured the decadal evolution of the properties. For the future, differences between the two scenarios (SSP5-8.5 and SSP2-4.5) are pronounced after 2030 (Figure 8). For the high scenario the C<sub>T</sub> concentrations reaches 2060 µmol.kg<sup>-1</sup> in 2040 and pH<sub>T</sub> is as low as 8. In both scenarios, the carbonate ion concentrations dropped below 200 µmol.kg<sup>-1</sup> in 2028. As noted above, the aragonite saturation state based on observations was 3.47 in July 2014 (Figure 7c and blue symbol in Figure 8d) and the lowest  $\Omega_{ar}$  value of 3.23 occurred in August-September 2023 (from the FFNN model, Figure 7c and 8d). The same is estimated in the projection (Figure 7d), close to the critical threshold of  $\Omega_{ar}$  = 3.25 for tropical coral reefs.

Figure 8: Time-series of (a) atmospheric and oceanic  $fCO_2$ , (b)  $C_T$  concentrations, (c)  $pH_T$  and (d)  $\Omega_{ar}$  in the southern Mozambique Channel based on a reconstruction for August for two scenario (SSP85, black line, SSP45 grey lines). Averaged observations (all seasons, July in blue) and the FFNN-LSCE model over 1985-2023 in August (orange) are also shown.

As of January 2025, the atmospheric  $CO_2$  is 426 ppm and 450 ppm should be reached in 2030 in the high scenario SSP5-8.5 (Figure 8a). In a global ocean model it has been suggested that at 450 ppm  $\Omega_{ar}$  would be around 3 in the South Indian Ocean and the Mozambique Channel, against 4 for pre-industrial  $\Omega_{ar}$  (Figure 4 in Hoegh-Guldberg et al., 2007). In our simulation, at 450 ppm,  $\Omega_{ar}$  is equal to 3.13 against 3.8 based on observations in May 1963. Extrapolating our result back in time, we estimated pH<sub>T</sub> at 8.18 and  $\Omega_{ar}$  equal to 4 at 280 ppm, close to the pre-industrial value estimated from dedicated reconstructions (Lo Monaco et al, 2021) or in global ocean models (Hoegh-Guldberg et al. 2007; Jiang et al, 2023).

Our calculation suggests that for a high emission scenario a risky level for corals ( $\Omega_{ar}$  < 3, Hoegh-Guldberg et al., 2007) could be reached as soon as year 2034, i.e. in the next 10 years. This calls for maintaining regular carbonate system observations in this region, if possible at all seasons, in order to follow at best their evolution and the potential impact on the channel ecosystem and especially coral reefs in the context of global warming and acidification that will dramatically persist in the near future.

## 3.5 A large anomaly observed in 2025

As mentioned above, fCO<sub>2</sub> data are relatively sparse in the Mozambique Channel and should be obtained for all seasons. To complete the shipboard observations, biogeochemical (BGC) Argo floats have been developed and successfully used for 10 years for air-sea CO<sub>2</sub> flux estimates and/or acidification especially in the Southern Ocean in the frame of the SOCCOM project (e.g. Sarmiento et al, 2023; Mazloff et al, 2023; Metzl et al, 2025d). The floats record profiles down to 1000 or 2000 m at a 10-day frequency. In November 2024, a BGC-Argo float (WMO ID 7902123) was launched in the Mozambique Channel at 38.51°E/22.65°S (last profile used here recorded on 4th May 2025). The  $fCO_2$  and  $\Omega$ ar from the  $pH_T$  float data were calculated using CO2sys as for the shipboard data (section 2.3). Interestingly, the float recorded high temperature in January 2025 (up to 29.8°C, Figure 9a), a signal probably linked to a MHW (Figure S10) that occurred at high frequency in this region (Mawren et al, 2022). Sea surface temperature from reanalysis products suggests a temperature as high as 31°C in this region in January 2025 (Figure S10). Consequently, the sea surface fCO<sub>2</sub> derived from the float pH<sub>T</sub> data reached values above 480 μatm (Figure 9b). This leads to a strong CO<sub>2</sub> source anomaly, in line with CO<sub>2</sub> fluxes anomalies associated to MHW in the South Indian subtropics (e.g. Li et al, 2024). The lowest pH<sub>T</sub> of 7.95 was recorded on 11<sup>th</sup> January 2025, i.e. lower than the pH<sub>T</sub> derived from the FFNN model (Figure 7b). The aragonite saturation state derived from the BGC-Argo data reached 3.2 in March 2025, the same as that which we estimate in 2025 from the simulation (Figure 8d). The observations from the BGC-Argo offer important information to complement the shipboard data and should be used along with SOCAT data to test constraint data-based products.

Figure 9: Time-series of (a) SST and  $pH_T$ , and (b)  $fCO_2$  and  $\Omega_{ar}$  in the southern Mozambique Channel based on BGC-Argo data (WMO7902123) in November 2024 to May 2025 (location of data in the insert map). Data from <a href="https://www.mbari.org/products/data-repository/">https://www.mbari.org/products/data-repository/</a>, last access 9/5/2025.

# 4. Summary and concluding remarks

New observations in 2021 and 2022 and historical fCO $_2$  data available since 1963 in the Mozambique Channel were used to evaluate the decadal trends of the carbonate system in this region. With adapted A $_T$ /S relationship for this region, we calculated C $_T$  concentrations, pH $_T$  and  $\Omega_{ar}$ . This calculation is first validated with in-situ A $_T$  and C $_T$  measurements obtained in January 2004, April 2019 and January 2021. Based on the data in January 2004 and 2021, we found a pH $_T$  decrease of -0.032 (using fCO $_2$  data) and -0.040 (using A $_T$  C $_T$  data) over 17 years. Because the seasonality is large, the decadal trends based on fCO $_2$  observations in 1963-2022 are evaluated for one season only (April-May). A FFNN model that reconstructed the monthly fields of the carbonate system is also used to investigate the trends for all seasons, but restricted to the period 1985-2023.

In this region where the ocean is a permanent  $CO_2$  sink of -0.25 (±0.06) molC.m<sup>-2</sup>.yr<sup>-1</sup>, fCO<sub>2</sub> observations available in April-May translate an acceleration of the acidification ranging from -0.012 decade<sup>-1</sup> in 1963-1995 to -0.027 (±0.003) decade<sup>-1</sup> in 1995-2022. Result from the FFNN model for all seasons suggest smaller pH<sub>T</sub> trends but, like in the observations, the decrease of pH<sub>T</sub> was faster in recent decades, -0.011 decade<sup>-1</sup> over 1985-1995 and -0.018 decade<sup>-1</sup> over 1995-2022. The FFNN model also suggests a faster trend in austral winter when the ocean  $CO_2$  sink is stronger and when the aragonite saturation state ( $\Omega_{ar}$ ) is low. In May 2022 we estimated  $\Omega_{ar}$  = 3.49, about 0.3 lower than observed in May 1963 ( $\Omega_{ar}$  = 3.86). The lowest  $\Omega_{ar}$  value of 3.23 was evaluated from the FFNN model in September 2023 that corresponds to the potential critical threshold value (3.25) for net reef accretion (Hoegh-Gulberg et al 2007) and could conduct to net reef dissolution (Eyre et al. 2018; Tribollet et al. 2019; Cornwall et al 2021).

A simple reconstruction and projection of the  $C_T$  concentrations based on anthropogenic  $CO_2$  in subsurface water and emissions scenario, suggests that the aragonite saturation state could be as low as 3 in the next 10 years. Following a previous work (Lo Monaco et al, 2021), the new data

presented here clearly reveal the progressive acidification in the Mozambique Channel and its acceleration in the recent decade with potential impacts on ecosystem including corals reefs areas like at Europa and Bassa de India. In a context where there is no sign of a slowdown in anthropogenic emissions, this already obvious acidification is alarming for the ocean health (Gattuso et al, 2015) and potential feedback on the ocean carbon cycle in general (e.g. Barrett et al, 2025). Understanding and quantifying the future response of phytoplanktonic and reef species in the context of global warming and acidification calls for adapted ocean biogeochemical models (Cornwall et al 2021) to take into account dynamics of bioerosion processes (see Schönberg et al 2017). In the Mozambique Channel, observations are still very sparse and the new observations presented here, including recent BGC-Agro data, offer important information to validate regional and global biogeochemical models that are not yet able to simulate correctly the seasonal cycle and decadal variability of the oceanic carbonate system. We strongly claim for maintaining regular sampling of ocean carbonate system parameters to reduce the model uncertainties and for adapted strategies at both scientific and political actions in the future.

#### Data availability:

762

765

770

777778

784

786

Data used in this study are available in SOCAT (Bakker et al, 2016, www.socat.info) for fCO<sub>2</sub> surface data, in GLODAP (Lauvset et al, 2022a, 2022b, www.glodap.info) for water-column data. The  $A_T$  and C<sub>T</sub> underway data from OISO and CLIM-EPARSES cruises are available in Seanoe (Metzl et al. 2025c, https://www.seanoe.org, https://doi.org/10.17882/95414 and https://doi.org/10.17882/102337). The BGC-Argo float data with derived carbon parameters are available from SOCCOM (https://soccom.org, https://www.mbari.org/products/data-repository/). The CMEMS-LSCE-FFNN are available at E.U. Copernicus Marine Service (https://resources.marine.copernicus.eu/products, https://doi.org/10.48670/moi-00047). The Mixed layer depth data publicly available on **CMEMS** are the website: https://data.marine.copernicus.eu/product/ (Multi Observation Global Ocean ARMOR3D L4 MULTIOBS\_GLO\_PHY\_TSUV\_3D\_MYNRT\_015\_012).

## **Authors contributions:**

CLM and NM are co-investigators of the ongoing OISO project. AT was chief scientist of the CLIM-EPARSES 2019 cruise and JFT was chief scientist of the RESILIENCE cruise. MG and FC developed the CMEMS-LSCE-FFNN model and provided the model results. NM started the analysis, wrote the draft of the manuscript and prepared the figures with contributions from all co-authors.

**Competing interest:** The authors declare that they have no conflict of interest.

Acknowledgments: The OISO program was supported by the French institutes INSU (Institut National des Sciences de l'Univers) and IPEV (Institut Polaire Paul-Emile Victor), OSU Ecce-Terra (at Sorbonne Université), the French programs SOERE/Great-Gases and ICOS-France. We thank the French Oceanographic for financial and logistic support the OISO program (https://campagnes.flotteoceanographique.fr/series/228/). **CLIM-EPARSES** The supported by TAAF (Terres Australes et Antarctic Françaises), IRD (Institut de Recherche pour le Développement), Fondation Prince Albert II de Monaco (www.fpa2.org), INSU (Institut National des Sciences de l'Univers), CNRS (Centre National de Recherche Scientifique), Museum National d'Histoire Naturelle (MNHN), UMRs LOCEAN-IPSL, ENTROPIE and LSCE. The RESILIENCE cruise (https://doi.org/10.17600/18001917) was supported by the French National Oceanographic Fleet, by the Belmont Forum Ocean Front Change project, by the ISblue project and by the French National program LEFE (Les Enveloppes Fluides et l'Environnement). We thank the captains and crew of R.R.V. Marion-Dufresne and the staff at IFREMER, GENAVIR and IPEV. We also thank Guillaume Barut, Jonathan Fin and Claude Mignon for their help during the OISO, CLIM-EPARSES and RESILIENCE cruises. The development of the neural network model benefited from funding by the French INSU-GMMC project "PPR-Green-Grog (grant no 5-DS-PPR-GGREOG), the EU H2020 project AtlantOS (grant no 633211), as well as through the Copernicus Marine Environment Monitoring Service (project 83-CMEMS-TAC-MOB). We acknowledge the Southern Ocean Carbon and Climate Observations and Modeling (SOCCOM) Project funded by the National Science Foundation, Division of Polar Programs (NSF PLR-1425989 and OPP-1936222). We thank all colleagues who contributed to the quality control of ocean data made available through GLODAP (www.glodap.info). The Surface Ocean CO<sub>2</sub> Atlas (SOCAT, www.socat.info) is an international effort, endorsed by the International Ocean Carbon Coordination Project (IOCCP), the Surface Ocean Lower Atmosphere Study (SOLAS) to deliver a uniformly quality-controlled surface ocean CO2 database. We thank Hermann Bange associate editor, and anonymous reviewers for their positive reports and comments that helped revising the manuscript.

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

- 6.