# Peer review of "Title: New observations confirm the progressive acidification in the Mozambique Channel"

_EGUsphere, 2025_

## Author Comment (AC1)

;;;;;;;;;;;; Review 1, 5/9/25

This study provides novel results on OA in the Mozambique Channel, addressing relevant scientific questions and contributing to the understanding of Global Change impacts in the Indian Ocean. The authors present recent observations from this region, contrasted with existing datasets and supported by climatologies and neural network approaches. The study design, concepts, methods, and data employed give the manuscript appropriate scientific significance.

However, several aspects require major revision to improve the quality of the research and the presentation of the results and conclusions. In particular, the manuscript would benefit from a clearer description of the study area and its main oceanographic features, a more detailed and structured explanation of the applied methodology, and a careful revision of the trend calculations. Addressing this last point is crucial to strengthen the main conclusions of the study.

AR-01: We thank the reviewer for her/his positive and clear report. Our responses are in blue.

Below, I provide my major and minor concerns.

Major comments

The introduction is concise and well-structured. However, it would be convenient to make explicit reference to the study area and include a description of the most characteristic processes that may act as sources of variability for the CO2 system. Alternatively, a short description could be included in the introduction, with further details provided in a Study Area subsection within the methodology.

AR-02: In the introduction we have referenced to the study area. In short to introduce our study we recalled OA in the global ocean, then in the Indian Ocean and the Mozambique Channel. We are not sure what specific information could be added regarding the processes. At the end of this section we suggest to add: "Both studies concluded that strengthening of acidification trend was mainly driven by ocean CO2 uptake."

The color code in Figure 1 does not allow interpretation. It is difficult to distinguish the tracks of each cruise, particularly the red ones (from 2010 onwards, when most cruises took place). Please consider using a unique color for each cruise to improve readability.

AR-03: Figure 1 aimed at showing that data exist for different years and identifying locations of crossing when exist. As suggested, we have revised the figure with a different color code. As suggested by reviewer 2 we also add the location of coral reefs (here Bassas de India and Europa in a second map with cruises tracks).

Figure 1 revised: Left: Tracks of cruises in the Mozambique Channel in the SOCAT data-base, version v2024 (Bakker et al., 2016; 2024). This includes recent OISO-31 and RESILIENCE cruises in 2021 and 2022. Color code is for Year. Black circles identified the coral reefs locations. Right: Tracks of cruises near the coral reefs area. Figures produced with ODV (Schlitzer, 2018).

[Figure]

Line 128: How was SST measured? And salinity? What were the instrumental uncertainties? How was the equilibrator temperature corrected relative to in situ values? Even if this information has been described in previous works, it would be helpful to include it in Section 2.2 so that readers can fully understand the methodology.

AR-04: As noted by the reviewer, the methods have been described in many previous studies and we thought it was not needed to recall again these information. Detailed information is available for each cruise in the Metadata file on-line in SOCAT. However, as suggested, we have added information on SST and salinity and mean of Teq-SST for the cruises in section 2.2 as follows:
"The sea surface temperature (SST) and equilibrium temperature were measured using SBE21 and SBE38 probes (accuracy 0.002°C) respectively. During the RESILIENCE cruise the difference of SST and equilibrium temperature was on average +0.088 ±0.066 °C (n= 6416). For all cruises, the sea surface salinity (measured with SBE21) was regularly checked with discrete samples and has been corrected if some drift was observed."

Lines 299–300: "…the climatology (Fay et al., 2024) or the FFNN model (Chau et al., 2024) is coherent compared to the data." Was any intercomparison test performed? It would be appropriate to include results of such an intercomparison, for instance, the mean difference between climatology/NN outputs and observations, and/or provide an additional figure in the Supplementary Material (e.g. time in months on the X-axis and difference on the Y-axis).

AR-05: Thank you. We have compared the values and added a figure in the Supp. Mat. as suggested

New Figure S1: Time-series of fCO$_2$ (µatm) and pH$_T$ in the southern Mozambique Channel based on observations (black circles) and from the FFNN model (grey diamonds) for the same periods. Standard-deviations are indicated by vertical bars. The differences (FFNN minus Observation) are also shown (Open squares, right axis). In 2018 the fCO$_2$ from the model is high compared to the observations.

[Figure]

Figure 3: The criteria for selecting these specific periods are not clear, and this choice may introduce bias. The averaging of the 2003, 2004, and 2014 cruises seems to give greater influence to the earlier years. It would be more informative to present the direct observations of each cruise with distinct markers. This approach would facilitate interpretation and enable clearer comparison with FFNN and climatology results. Including error bars would also strengthen the figure.

AR-06: Based on your comment and reviewer 2 we have revised both Figures 3 and 4

Figure 3 revised

"Figure 3: Seasonal cycle of (a) fCO$_2$ (µatm) and (b) pH$_T$ in the southern Mozambique Channel (24-30°S). Average observations are presented for each cruise (colored circles). The full seasonal cycles are shown for the monthly climatology (reference year 2010, Fay et al, 2024) and for the FFNN model for years 2010 and 2022 with respective error bars."

[Figure]

Figure 4 revised

"Figure 4: Seasonal cycle of C$_T$ (µmol/kg) in the southern Mozambique Channel (24-30°S). Average observations are presented for each cruise (colored circles). The full seasonal cycles are shown based on the monthly climatology for a reference year 2010 (Fay et al, 2024) and the FFNN-LSCE model for year 2010 (Chau et al, 2024). The mixed-layer depth (MLD in m, blue line) is averaged in this region (from multi-year reprocessed monthly data, ARMOR3D L4, https://doi.org/10.48670/moi-00052, last access 20/4/2025)."

[Figure]

Section 3.3 does not appear to align well with the overall structure of the paper. Although the main result derived from this section is interesting for the broader discussion, it does not directly fit within the objectives. I therefore suggest moving this section to an Appendix and including the figure as Supplementary Material.

AR-07: We are not sure to understand this comment. Note that as suggested by Reviewer 2 the title of Section 3 has been changed to: "3 Results and discussion"

Section 3.3.2: How were the trends calculated? Were they derived from observational data or from climatology/NN outputs? This is a critical point that needs clarification. For example, between 1963 and 1995 (line 498) a trend is reported, but only two data points exist. In this case, it would be more appropriate to report the change between 1963 and 1995 (-0.040 units) rather than describing it as an interannual trend, which would be biased without intermediate data. The same concern applies to trends reported for 1995–2019 (lines 507–509) and 1995–2022 (lines 514–515). Similarly, in Table 4 the limited number of observations in the second half of the 20th century (only in 1963 and 1995) introduces biases in trend estimation. In addition, the high variability observed since 2018 could influence the calculated trends. Under these circumstances, no interannual or decadal trends can be identified with any statistical significance; instead, it would be more reliable to compare recent values with those from 1995 to estimate the magnitude of change. These changes will require a revision of the main conclusions of the paper, as well as the abstract

AR-08: We agree with this comment. Text revised on line 498.
AR-08: For the line 507-515 we recall results from Lo Monaco et al 2021) and Chakraborty et al, (2024) No change.
AR-08: This is correct, for the first period, observations are only available in 1963 and 1995. However, we listed all trends deduced from observations or the FFNN model in Table 4. The conclusions of the results for the decadal changes are mainly based on the trends deduced from the FFNN model but we think it is useful to indicate what we learn from observations even if this is only between 2 periods.

Section 3.4 (line 622): The calculation applied in this section may be difficult for readers to follow. The reconstruction of past and future values and the data sources used are not entirely clear. It might be beneficial to expand on the methodological details, either in the methodology section or in a dedicated Appendix, and limit the current section to the discussion of results. Additionally, please clarify the estimation error associated with Eq. 2.

AR-09: The methodology was described in previous work (Metzl et al, 2025b), and we thought it was not useful to recall the details. However, as suggested we have added this information in the Supp. Mat.
The error associated to Eq 2 was somehow indicated on line 625: +0.512 ±0.050 µmol kg$^{-1}$ µatm$^{-1}$
Equation 2 presented the way we calculated $C_T$ for each time step based on the $C_{ant}$ annual rate.

Added in Supp. Mat. new figure:

Figure S8: The relationship between $C_{ant}$ and atmospheric $CO_2$ used for the reconstruction (Equation 2) was described by Metzl et al (2025b). It was evaluated from the $C_{ant}$ concentrations in subsurface using data in 1987 to 2021 and correlated to the atmospheric $CO_2$ concentrations. (a) Time-series of anthropogenic $CO_2$ concentrations ($C_{ant}$) over 1987-2021 estimated in subsurface (layer 100-150m) from the GLODAP-v2023 data (Lauvset et al, 2024) completed with OISO cruise in 2021 (location of selected stations in the insert map, color code is for year). The figure shows the $C_{ant}$ concentrations calculated for each sample (black) and the $C_{ant}$ averaged in the layer 100-150m for each period (grey triangles). Over the period 1987-2021, the Cant trend is +1.03 ±0.14 µmol kg$^{-1}$ yr$^{-1}$ (dashed grey line). The red curve is the atmospheric $fCO_2$. (b): same data for $C_{ant}$ versus atmospheric $fCO_2$ (slope= +0.512 ±0.050 µmol kg$^{-1}$ µatm$^{-1}$).

[Figure]

Minor comments

Lines 41–42: Is this referring to surface measurements? Please clarify.

AR-10: Thank you, yes this is for surface. Lines revised as follows: "the sea surface pH could decrease by 0.4 and aragonite saturation state (Ωar) could be as low as 3 in the tropics by 2100"

Line 47: Replace "fugacity of CO2" with "CO2 fugacity".

AR-11: Thank you, corrected

Lines 47–50: The inclusion of these trends may appear arbitrary and potentially confuse the reader, since the study region has not yet been specified in the introduction. While the Indian Ocean

reference is understandable, the criteria for also including the North Pacific is unclear. As a suggestion, it may be more relevant to mention the decrease in pH using trends from time-series stations (Bates et al., 2014), noting that most are located in the Northern Hemisphere and that knowledge of OA in the Southern Hemisphere remains limited. This could help emphasize the importance of the paper and engage the reader.

AR-12: In the introduction we have selected the range of pH trends previously quantified (slow in the North Pacific and rapid in the Indian Ocean). Here we refer to the results from $fCO_2$ data (SOCAT), whereas Bates et al (2014) synthetized results derived from $A_T$-$C_T$ data at several time-series. No change.

Line 49: What does "TS.decade-1" mean? Please use "units decade-1" or simply "decade-1" instead, and apply this consistently throughout.

AR-13: Thank you. We have changed the unit and pH to $pH_T$ in the text, tables and figures.

Line 58: Acceleration with respect to …?

AR-14: corrected as follows: "… it has been shown that the Mozambique Channel experienced an acceleration with respect to the acidification in recent years".

Line 135: Were only three gases used? Was a 0 ppm gas not included to zero and span the system (Pierrot et al., 2009)? If not, how were the xCO2 measurements corrected?

AR-15: Yes, three gases were used during our cruises. This is what has been done for years during OISO cruises and validated through the quality control in SOCAT. In this manuscript we do not need to detail the instrumental and corrections. No change.

Lines 160–162: Please include statistical information (e.g., RMSE and r2).

AR-16: Thank you, error statistics and R2 added:
$A_T$ ($\mu$mol.kg$^{-1}$) = 73.841 ($\pm$ 1.15) * SSS − 291.02 ($\pm$ 40.4) (n= 548, $r^2$= 0.88).

Lines 193–194: "For pH, the decrease of -0.005 over three years, i.e., -0.0017 yr-1, is surprisingly close to what is generally observed at global scale and over several decades (-0.017 $\pm$ 0.004 per decade)." Please use consistent units when reporting both trends.

AR-17: Thank you. Units changed.

Table 2: Why do the mean SST and SSS values for CLIM-EPARSES fCO2 and CLIM-EPARSES AT-CT not match? They should be identical if only the computation of the carbon system variables differs. The same issue appears in Table 3 with OISO-11 and OISO-31. Please clarify.

AR-18: The mean SST and SSS were evaluated from the data points available for each instrument. For $fCO_2$ we recorded one data per 5 minutes, for $A_T$ and $C_T$ one data per 20 minutes. This is why we indicated the number of data for each method (294 for fCO2 and 70 for AT-CT). The same applied for Table 3. No Change.

Line 316: Please specify the salinity to which CT is normalized.

AR-19: $C_T$ is normalized at salinity 35. Information added as follows: "normalized $C_T$ at salinity 35"

Line 323: Is there a reference supporting the reported trend?

AR-20: Yes, the SST trend of +0.11 ± 0.009 °C per decade since the 1960s was evaluated from data extracted at http://iridl.ldeo.columbia.edu/SOURCES/.NOAA/.NCDC/.ERSST/.version5/.anom/, last access 12 May 2025.

We suggest add a figure in the Supp. Mat.:

Figure R1: Monthly sea surface temperature anomalies (°C) at 25°S-40°E obtained from http://iridl.ldeo.columbia.edu/SOURCES/.NOAA/.NCDC/.ERSST/.version5/.anom/, last access 12 May 2025. The red line is the linear trend of +0.011 °C per year (i.e. +0.11 °C per decade).

[Figure]

Lines 365–366: This statement would be more appropriate in the Results section.

AR-21: Thank you. This was listed in the figure caption: "CT concentrations are high when MLD is deeper in austral winter." Now moved in the Result section as follows:
"The progressive $C_T$ increase of about +30 µmol.kg$^{-1}$ from March to August is likely driven by vertical mixing when MLD is deeper in austral winter (Figure 4)."

Line 387: Consider replacing "increased" with "reinforced".

AR-22: We agree with this suggestion and have corrected: "The FFNN model also suggests that the sink reinforced over 2016-2021 with a perceptible faster increase of $C_T$ (Figure S3)."

Lines 424–425: How were eddies identified? Were satellite images used, or is there a reference?

Lines 424–425: "We noticed that in 2021, the properties present a high variability along the track linked to the presence of eddies".

AR-23: As the distribution in 2021 was much more variable than in 2004 (Figure 6), we suggested that the spatial variability in 2021 was linked to the presence of eddies. This was identified from SST and SSS data as well as from reanalysis (Figures R2 and R3, not included in the Manuscript).

Figure R2: Distribution of SST and SSS along the same track in January 2004 (black symbols) and January 2021 (grey symbols). In 2021, the variability of salinity suggested the presence of eddies (see also Figure R3).

[Figure]

Figure R3: Map of geopotential height in January 2021 (left) and January 2004 (right) highlighting an eddy structure (yellow circle) in 2021. Maps constructed from CMEMS: dataset-armor-3d-rep-monthly ( https://doi.org/10.48670/moi-00052), last access 11 Sept. 2025.

[Figure]

Line 428: ΔfCO2 should be defined as the difference between oceanic and atmospheric fCO2 before it is mentioned for the first time.

AR-24: Thank you, ΔfCO2 now defined here for the first time (on line 428): $fCO_2^{ocean}$-$fCO_2^{atm}$ = ΔfCO2 = -0.04 ±3.11 µatm.

Line 496: What does "difference of ΔfCO2" mean? Please clarify.

AR-25: This was an unclear repetition and we have deleted "difference".
Line 496 revised: "Back in the 1960s, the observations in 1963 indicate that the ocean was a CO2 sink in May (Figure 7a), the value of ΔfCO2 = -32.2 µatm being almost the same as observed in May 2022 (ΔfCO2 = -32.5 µam)."

Line 500: Please specify the units.

AR-26: No unit needed but line 500 "Over 32 years this pH change was driven by the CT increase (effect on pH= -0.045), the AT increase (+0.012) and the warming of 0.95°C (-0.015)."
Revised as follows:
"Over 32 years this pH change was driven by the $C_T$ increase (effect on pH= -0.045), the $A_T$ increase (effect on pH= +0.012) and the warming of 0.95°C (effect on pH=-0.015)."

Line 639: Instead of simply stating "compared well", it would be helpful to report the mean differences between this method and the observations, as well as between this method and the FFNN estimates/climatology data.

AR-27: Line 639: "The reconstructed C$_T$, fCO$_2$, pH and Ωar for August compared well with the observations (in July) and with the FFNN model in August (Figure 8) indicating that the simulation captured the decadal evolution of the properties".

We agree that "compared well" as visualized in Figure 8 need more detail.
We have reported the differences corresponding to the results in Figure 8 (Table R1 and Figure R4) and suggest add the table R1 and Figure R4 in the Supp. Mat.
* * *
Table R1: Mean difference between the reconstruction and the FFNN model for August 1985-2022 and with observation in July 2014. SD in brackets.
* * *
| Method | Year | fCO2 µatm | CT µmol/kg | pH TS | War nu |
|---|---|---|---|---|---|
| Sim-FFNN | 1985-2022 | 2.6 (4.9) | 7.9 (3.4) | -0.003 (0.005) | -0.005 (0.029) |
| Sim.-Obs. | 2014 | 2.8 | 4.8 | -0.002 | -0.099 |
* * *
Figure R4: Time-series of the difference of (top) oceanic fCO2 and CT concentrations and (bottom) pH and Ωar between the reconstruction using SSP85 scenario and the FFNN-LSCE model over 1985-2023 in August or with observations (July 2014, red). The differences are calculated from data presented in Figure 8.

[Figure]

Lines 679–680: Please provide a reference.

AR-28:  Lines 679–680: "Our calculation suggests that for a high emission scenario a risky level for corals ($\Omega$ar < 3) could be reached as soon as year 2034, i.e. in the next 10 years."
We understand that the reviewer asks for a reference regarding the "risky level". As indicated in the manuscript (lines 43 and 562), we refer to Hoegh-Guldberg et al., (2007).
Sentence revised as follows:  "Our calculation suggests that for a high emission scenario a risky level for corals ($\Omega$ar < 3, Hoegh-Guldberg et al., 2007) could be reached as soon as year 2034, i.e. in the next 10 years.

Figure 9: Is it possible to add error bars?

AR-29: Figure 9 present the data from the BGC-Argo float select in surface waters (two data point at 7 and 11m) for each period. No need to add error bars on this figure.

;;;;;;;; end response to reviewer 1

---

## Author Comment (AC2)

Metzl et al. compile observations and observation-based reconstructed surface ocean carbonate parameters to assess historical and future trends in ocean acidification in Mozambique Channel. They find an acceleration of acidification since the 1990s and predict the timing of the crossing of critical thresholds for coral reefs. Given the scarcity of observations in this region, and the Indian Ocean in general, as well as the existence of nearby coral reef ecosystems, this is an important contribution to the scientific literature. However, there are several issues that should be addressed in order to clarify the methodology, reporting, and implications of the results.

AR-01: We thank the reviewer for her/his positive and clear report. Our responses are in blue.

Seasonality of the observations and reconstructions:

Given the significant seasonality of surface ocean carbonate chemistry in this region (Figures 3 and 4) and the desire to report trends without adequate observational coverage in all seasons, it would be useful to better align trend comparisons in the same month(s). In section 3.3.2, trends from historical observations from April-May should be compared to trends from the FFNN product from April-May rather than only April. Figure 3 shows there are significant differences in the climatology between April and May. In addition, it is unclear why August is used for the future predictions (Line 636) when the comparison is being made to July observations (639). It would be better to align those FFNN months with the available observing months or report seasonal FFNN results that encompass the observations (Jan-Mar, Apr-Jun, Jul-Sep).

AR-02: We understand the question regarding the selection of months for the trend analysis. The reviewer is correct; we first described the seasonality in order to select data for the trends described in section 3.3.2 and for the observations we select cruises in April-May. We have calculated the trends in April and May separately and found the same results. This is confirmed when exploring the trend of the differences (Figure R1). We suggest add the results for May in Table 2 to show that the same trends are obtained for both months.

Table R1: Same as Table 4 after adding results from FFNN in May.
* * *
Trends of properties in the southern Mozambique Channel derived from observations and the FFNN model. For observations, the trends are evaluated for April-May season only. For FFNN, trends are estimated for all seasons or only for January, April, May and July. Standard-deviations are in bracket.

| Method | Period | $fCO_2$ $\mu atm.yr^{-1}$ | $C_T$ $\mu mol.kg.yr^{-1}$ | $A_T$ | pH $TS.yr^{-1}$ |
|---|---|---|---|---|---|
| Obs April-May | 1963-1995 | 1.11 | 0.91 | 0.52 | -0.0012 |
| Obs April-May | 1963-2022 | 1.84 (0.21) | 0.69 (0.20) | 0.08 (0.13) | -0.0020 (0.0002) |
| Obs April-May | 1995-2022 | 2.57 (0.30) | 0.49 (0.52) | -0.34 (0.22) | -0.0027 (0.0003) |
| FFNN April | 1985-2023 | 1.74 (0.03) | 1.01 (0.07) | 0.03 (0.07) | -0.0017 (0.0000) |
| FFNN May | 1985-2023 | 1.71 (0.03) | 1.07 (0.05) | 0.07 (0.05) | -0.00167 (0.00003) |

Figure R1: Time-series of the difference between May and April for oceanic $fCO_2$ and $C_T$ concentrations from the FFNN-LSCE model over 1985-2023. There is no trend in these differences. Consequently the $fCO_2$ and $C_T$ trends for April or May are the same (Table R1).

[Figure]

AR-02: Concerning the month selected for prediction (August), we select August as this is the month with the lowest $\Omega$ar. Unfortunately, in austral winter there are observations only in July and we think is was useful to add this data point on the figure. Note that Reviewer 1 asked for detail comparison between reconstruction and FFNN results and our response is copied below:

Line 639: "The reconstructed CT, fCO2, pH and $\Omega$ar for August compared well with the observations (in July) and with the FFNN model in August (Figure 8) indicating that the simulation captured the decadal evolution of the properties".

We agree that "compared well" as visualized in Figure 8 need more detail.
We have reported the differences corresponding to the results in Figure 8 (Table R1 and Figure R4) and suggest add the table R1 and Figure R4 in Supp Mat.
* * *
Table R1: Mean difference between simulation and FFNN for August 1985-2022 and with observation in July 2014. SD in brackets.
* * *
| Method | Year | fCO2 µatm | CT µmol/kg | pH TS | War nu |
|---|---|---|---|---|---|
| Sim-FFNN | 1985-2022 | 2.6 (4.9) | 7.9 (3.4) | -0.003 (0.005) | -0.005 (0.029) |
| Sim.-Obs. | 2014 | 2.8 | 4.8 | -0.002 | -0.099 |
* * *
Figure R4: Time-series of the difference of (top) oceanic fCO2 and CT concentrations and (bottom) pH and Ωar between the reconstruction using SSP85 scenario and the FFNN-LSCE model over 1985-2023 in August or with observations (July 2014, red). The differences are calculated from data presented in Figure 8.

[Figure]

In addition, it is not always clear what time period is being referred to. For example, is the FFNN trend referred to in Lines 501-502 FFFN annual or FFNN April?

AR-03: Thank you. We have specified the periods more clearly when appropriate: Lines 501-502 revised as follows:

"In contrast, the neural network suggested smaller pH trends. However, as in the observations, the annual pH change from the model was faster in recent decades (-0.0018 yr$^{-1}$ over 1995-2022 against -0.0011 yr$^{-1}$ over 1985-1995, Table 4)."

Coral reef implications:

How close in space are the historical sampling locations and the coverage of the FFNN reconstructions to the coral reefs in the Mozambique Channel? It is not obvious whether surface ocean carbonate chemistry presented here overlaps at all with nearshore subsurface coral reefs. More evidence is needed to conclude that the current and projected low aragonite saturation state conditions presented here are likely to be the same conditions occurring near coral reefs in this region. In Figure 1, it would be useful to include the extent of coverage used from the FFNN products as well as locations of coral reef ecosystems. The shading in Figure 1 needs to be described in the caption, and if they represent bathymetry, a scale for that should also be included.

AR-04: This is an important point. In our present analysis, we explored the ocean acidification in the open ocean waters from available data in the Mozambique Channel. During few cruises, CLIM-EPARSES 2019 and RESILIENCE 2022 observations were also conducted very close or within the coral

reefs (Europa at 22.5°S-40.5°E, Bassas de India at 22°S-39.5°E). With only two cruises, these data could not support the trend analysis. They are available in SOCAT and Seanoe (as listed in the Data availability section) and might be used for future investigation with new observations conducted in the same region.

As shown in Figure 2, in April 2019 the properties north of a front at 23°S and around 22.5°S were highly variable. This corresponds to observations conducted around Europa. Detailed information is shown in Figure R2. Around Europa, we observed a large spatial and diurnal variability making it challenging to detect a long-term trend in these complex domains. In 4 days (8-11 April) the $fCO_2$ ranged between 384 and 398 µatm and pH ranged between 8.040 and 8.052, i.e. 0.012. This is a large signal compared to the mean difference observed over 3 years in the band 23-26°S (Table 2 of the manuscript, +7.9 µatm for $fCO_2$ and -0.005 for pH). This is why we select the region south of 23°S for the trends.

In addition, although we used the 0.25x0.25 degree resolution version of the FFNN model, the model is still not adapted for meso or small scales reconstructions (work in progress for coastal zones). Further analysis is needed to explore the distribution in corals reefs areas in detail; this will be prepared in other studies along with reconstructions from coral cores sampled in the reefs (e.g. Alaguarda et al, 2025; Alaguarda et al, accepted,)

On this issue, we have added in the results (section 3.1):
"Given the variability observed around Europa Island and the front identified at 22.5°S in April 2019 (Figure 2) the data were averaged in the band 23°-26°S."

Figure R2: pH versus $fCO_2$ observed in April 2019 around Europa (location of data in the insert map, and color code is for $fCO_2$). In 4 days (8-11 April) the pH ranged between 8.040 and 8.052, i.e. a change of 0.012 being more than twice compared to the difference between 2019 and 2022 (-0.005 for pH).

[Figure]

AR-05: As suggested, the color code for the time of cruises has been changed and locations of coral reef named in the manuscript also added in Figure 1 (circles for Bassas de India and Europa). We also added a map to show in detail the 2 locations with the bathymetry scale as suggested.

Figure 1 revised: Left: Tracks of cruises in the Mozambique Channel in the SOCAT data-base, version v2024 (Bakker et al., 2016; 2024). This includes recent OISO-31 and RESILIENCE cruises in 2021 and 2022. Color code is for Year. Black circles identified the coral reefs locations. Right: Tracks of cruises near the coral reefs area. Figures produced with ODV (Schlitzer, 2018).

[Figure]

Other comments:

Line 18: I assume "TS" refers to pH in total scale? If yes, define that acronym at first instance in the main text (line 49) and just say in the abstract "ranging from a pH change of -0.012 decade -1" in the abstract. It wasn't until I got to Figure 2 that I figured out what TS was.

AR-06: Yes thank you, TS refer to total scale. We have changed pH to $pH_T$ in the manuscript and tables, and units corrected.

Line 25: How do you define "low"? Should this be "lowest"?

AR-07: Thank you, changed by "the lowest" on line 25.

Line 27: Briefly describe the two emission scenarios here.

AR-08: Line 27 revised (in short for the abstract): "A projection of the $C_T$ concentrations based on observed anthropogenic $CO_2$ in subsurface water and future anthropogenic $CO_2$ emissions scenario, suggests that a risky level for corals ($\Omega$ar < 3) could be reached as soon as year 2034."

Line 160: Even though this alkalinity proxy is described in another contribution, the uncertainty in the alkalinity proxy should be provided here.

AR-09: The same comment by Reviewer 1. Error and R2 added in Eq1:
$A_T$ ($\mu$mol.kg$^{-1}$) = 73.841 (± 1.15) * SSS − 291.02 (± 40.4) (n= 548, r$^2$= 0.88) (Eq. 1)

Line 172: Similarly, a brief overview of the estimated uncertainty of reconstructed fCO2 should be provided here. Given one season has no data for training the neural network, are there any assessments of predicted uncertainty during that season?

AR-10: Recall Line 172: "A full description of the model is presented in Chau et al (2024)."
Each data point in the FFNN model is associated to an uncertainty value (in the CMEMS files) based on an ensemble of 100 reconstructions. For example, in the region investigated here, the mean error and standard-deviation of $fCO_2$ is 10.2 ±3.5 µatm for 24336 data points in 1985-2023 (see Figure R3). Here, we referred to the original publication and the reader can find the detail of the model, their constraints and uncertainties evaluated in specific regions and at global scale.

Line 172 has been revised as follows: "A full description of the model is presented in Chau et al (2024) and the datasets including uncertainties are available under the DOI https://doi.org/10.48670/moi-00047

Figure R3: Time-series of the of oceanic $fCO_2$ (top) and $fCO_2$ uncertainties (bottom) in the region 17-30°S around 40°E from the FFNN model.

[Figure]

Line 174: This section seems to be both Results and Discussion?

AR-11: This is correct, thank you. Title section revised: "Results and Discussion"

Lines 322-325: Given the rapid warming in this region, have the authors considered assessing the thermal and non-thermal components of the fCO2 trend? Later in the results, values are presented that correct for the SST warming, but the method used for doing this is not presented.

AR-12: The effect of warming was tested in a previous study (Lo Monaco et al, 2021). Here we discussed the effect of warming on pH changes between 1963 and 1995 (line 500). Note that based on reviewer 1 comment, we added a figure of the SST anomalies and lines 322-325 revised as follows:"Specifically, in the southern Mozambique Channel the SST has increased by +0.11 ± 0.009 °C per decade since the 1960s (Figure S4), a signal that should be taken into account when interpreting the decadal trends of carbonate properties and CO2 fluxes. In January 2025 the SST anomaly reached +1.6°C at 25°S in the Channel."

Figures 2 and 3: Should FFNN-2010 be presented as gray to match the observations from the same time period like FFNN-2022?

AR-13: Thank you, this is probably for Figures 3 and 4. FFNN-2010 line revised in grey.

Figure 3: Is FFNN-2022 missing?

AR-14: Thank you, this is probably for Figure 4. This figure aimed at showing the link between $C_T$ and MLD to explain the $C_T$ increased when MLD was deeper and highlight presence of anomalies. The results from FFNN model for 2022 added (Figure 4 revised below). This shows the $C_T$ increase between 2010 and 2022 and highlight anomalies observed in April 2018 and 2022, i.e. challenging to detect the $C_T$ seasonality from observations only.

Figure 4 revised:
"Figure 4: Seasonal cycle of $C_T$ ($\mu$mol kg$^{-1}$) in the southern Mozambique Channel (24-30°S). Average observations are presented for each cruise (colored circles). The full seasonal cycles are shown based on the monthly climatology for a reference year 2010 (Fay et al, 2024) and the FFNN-LSCE model for year 2010 (Chau et al, 2024). The mixed-layer depth (MLD in m, blue line) is averaged in this region (from multi-year reprocessed monthly data, ARMOR3D L4, https://doi.org/10.48670/moi-00052, last access 20/4/2025)."

[Figure]

Figure 3: Include an interpretation of what may be causing the anomalous observations in April 2018-2022. Is this due to the variability from the 2021 eddies described in Lines 424-425?

AR-15: Thank you, this is probably Figure 4. Figure 4 aimed at describing the $C_T$ seasonality derived from climatology and its link with the MLD. We added averaged observed data on this figure to show that, in this region, the full seasonal cycles cannot be derived from observations.

As noted in the figure captions the average observations were evaluated in the band 24-30°S. However, in April 2018 the data were available mostly at 30°S, whereas in April 2019 and 2022 at 24°-27°S. This explains in part that $C_T$ in 2018 was higher (orange circles in Figure 4). In addition, Figure 4 also shows that in March-April-May the MLD deepens; therefore, a small change of MLD from year to year could impact the surface properties leading to significant variability of $C_T$. The result of the FFNN model also shows that in 2022 the increase of $C_T$ from April to May could be as high as +18.4 $\mu$mol/kg, much larger than in 2010 (+6.8 $\mu$mol/kg). Figure R4 below (not in the manuscript) shows that the variations from April to May could be higher than 15 $\mu$mol/kg (in 2017, 2021 and 2022). Given these results, i.e. observations available for few years and the inter-annual variability, we have selected specific season for the trend analysis, the main aim of the study. Finally,

we note that the data from the BGC-Argo float also indicate large variability of $C_T$; specifically we observed that $C_T$ increase when MLD is deeper (Figure R5). The analysis of the inter-annual variability as depicted in figure R4 should be further studied using more observations, satellite data, MLD derived from model at small-scale, and extend the FFNN results to year 2025 (period of the BGC-Argo float).

Figure R4: Time-series of the difference between May and April for oceanic $fCO_2$ and $C_T$ concentrations from the FFNN-LSCE model over 1985-2023. This highlights the variability in 2017, 2021 and 2022.

[Figure]

Figure R5: Time-series of MLD and surface $C_T$ in the southern Mozambique Channel based on BGC-Argo data (WMO7902123) in January to May 2025. The large $C_T$ increase (> 15 µmol/kg) in February or from late March to late April occurred when the MLD was deeper (> 50m).

[Figure]

Lines 418-419: Measurement and calculated parameter errors should be stated somewhere. Are these errors a part of the standard deviations in Table 3?

AR-16: Standard deviations in Table 3 correspond to those of the mean values estimated for each cruise along the track (27-29°S/40-43°E). For the errors on parameters we suggest to add 2 lines in Table 3 to compare the differences in regard to the errors of measurement or calculations (noted as * in the table).

| Cruise Method Period | Nb | SST °C | SSS - | $A_T$ μmol.kg$^{-1}$ | $C_T$ | $fCO_2$ μatm | pH TS | Atm. $xCO_2$ ppm |
|---|---|---|---|---|---|---|---|---|
| **Difference 2021-2004** | | | | | | | | |
| Underway fCO2 | | 0.532 | 0.225 | 16.7 | 28.3 | 37.8 | -0.032 | 37.4 |
| Underway AT-CT | | 0.400 | 0.267 | 11.5 | 29.4 | 45.7 | -0.040 | |
| | | | | | | | | |
| Error using fCO2 | | 0.01 | 0.01 | 4 | 7.3 * | 2 | 0.014* | |
| Error using AT-CT | | 0.01 | 0.01 | 4 | 4 | 13.9* | 0.007* | |

Figure 6: Everything in the figure legend needs to be defined in the caption. Unclear what fCO2 and AT-CT are? Measured or calculated?

AR-17: In this plot we show results derived from underway fCO2 measurements or underway $A_T$ $C_T$ measurements.

Captions revised as follows: Figure 6: Distribution of measured or calculated $C_T$ (a, μmol kg$^{-1}$), AT (b, μmol kg$^{-1}$), $fCO_2$ (c, μatm) and pH (d, TS) along the same track in January 2004 (black symbols) and January 2021 (grey symbols). Values derived from $fCO_2$ measurements are in filled symbols/lines, those from the $A_T$ $C_T$ measurements in open symbols/dashed lines. In (c) the red lines represent the atmospheric $CO_2$ in 2004 and 2021. Average values and their differences are presented in Table 3.

Line 564: Unclear what is meant by "ambient conditions" here.

AR-18: Line 563-564: "Note that there are reefs know to thrive at $\Omega$ar < 3.25 but that their species composition and coral cover is different than at ambient conditions (e.g. Strahl et al. 2015 in Papua New Guinea; see also review by Camp et al. 2018)."

Than you, we clarified for "ambient conditions" as follows:
« Note that there are reefs know to thrive at $\Omega$ar <3.0 like at volcanic CO2 seeps in Papua New Guinea ($\Omega$ar = 2.41, Strahl et al. 2015; see also review by Camp et al. 2018) but that their species composition and coral cover are different than at ambient conditions (i.e. $\Omega$ar > 3.3 considering Hoegh-Gulberg et al. 2007). However, Strahl et al. (2015) showed that calcification rate seems to vary among coral species, suggesting take conclusions of Hoegh-Gulberg et al (2007) with caution".

Line 636: How is "low" defined? Is "lowest" meant here?

AR-19: Thank you. Line 636 revised: "To explore the change of the aragonite saturation state, we applied this model (Eq. 2) for August when $\Omega$ar is the lowest."

Line 696-697: Reference the methodology for deriving fCO2 from pH.

AR-20: Reference added: New line: "The $fCO_2$ and $\Omega ar$ from the pH float data were calculated using CO2sys as for the shipboard data (section 2.3)."

Figure 9b: Define "War" in legend.

AR-20: "War" is $\Omega ar$

Figure 744: Is it a permanent sink given the observations from 2025? What are the potential implications of increasing marine heatwaves in this region to the CO2 sink?

AR-20: Interesting question. The effect of the MHW on the $CO_2$ sink in this region would be analyzed in further studies, including estimates of the annual integrated $CO_2$ flux over the full domain (Mozambique Channel and SW Indian Ocean). We think this could be achieved using the FFNN model extended to year 2025 as that was tested recently at global scale for the year 2023 (Müller et al, 2025).

;;;;;;;;;;; Reference in the review not listed in the manuscript:

Alaguarda, D, Chevalier N, Klein N, Massé A, Tribollet A (accepted) Lipid biomarkers of microboring communities in living massive corals: an interesting tool for understanding their composition and abundance. Coral Reefs.

Alaguarda, D, Brajard J, Lguensat R, Tribollet A (2025) Machine learning approach to study microboring assemblage dynamics in two genera of massive corals. Limnology & Oceanography Methods. https://doi.org/10.1002/lom3.10714

Müller, J.D., Gruber, N., Schneuwly, A. et al. Unexpected decline in the ocean carbon sink under record-high sea surface temperatures in 2023. Nat. Clim. Chang. 15, 978–985 (2025). https://doi.org/10.1038/s41558-025-02380-4

;;;;;;;;;;; end responses Reviewer 2

---

## Author Response (AR2)

Response to Editor comment, 3 November 2025

Dear Nicolas Metzl and co-authors,

the revised version of your manuscript was reviewd again by one reviewer who recommends publication as is; however with one minor correction:

'As a final recommendation, I suggest adding a note in the caption of Table 4 to clarify that the reported trends are based on limited observations: as the authors state, these trends are useful and informative for understanding the changes, but they should not be considered conclusive due to the lack of sufficient data.'

So, I am pleased to let know that your maunscript is now acceptable for publication (with the small correction mentioned above).

Thank you for submitting your work to Biogeosciences.

With best regards Hermann Bange

Author response: We thank the reviewer and Hermann Bange for the positive report and the suggestion. We have added the information in the caption of Table 4.

Sincerely,

Nicolas Metzl